# Polycystic Ovary Syndrome, Subclinical Hypothyroidism, the Cut-Off Value of Thyroid Stimulating Hormone; Is There a Link? Findings of a Population-Based Study

**DOI:** 10.3390/diagnostics13020316

**Published:** 2023-01-15

**Authors:** Ehsan Rojhani, Maryam Rahmati, Faegheh Firouzi, Marzieh Saei Ghare Naz, Fereidoun Azizi, Fahimeh Ramezani Tehrani

**Affiliations:** 1Reproductive Endocrinology Research Center, Research Institute for Endocrine Sciences, Shahid Beheshti University of Medical Sciences, Tehran 1985717413, Iran; 2Tehran Medical Branch, Islamic Azad University, Tehran 19395/1495, Iran; 3Endocrine Research Center, Research Institute for Endocrine Sciences, Shahid Beheshti University of Medical Sciences, Tehran 1985717413, Iran

**Keywords:** polycystic ovary syndrome (PCOS), subclinical hypothyroidism (SCH), thyroid stimulating hormone (TSH), thyroid, prevalence

## Abstract

Despite solid evidence regarding the association of over-hypothyroidism with polycystic ovary syndrome (PCOS), the relationship between PCOS and subclinical hypothyroidism (SCH) is still a topic of debate. In the present population-based study, we aimed to assess if there is a difference between PCOS and the control group regarding the upper reference limit of thyroid stimulating hormone (TSH). We also aimed to identify the prevalence of SCH in women with PCOS compared to controls. This study was conducted on data collected in the Iranian PCOS prevalence study and the Khuzestan PCOS prevalence study. Participants that met our eligibility criteria were categorized into two groups: PCOS (*n* = 207) and control (*n* = 644). Quantile and logistic regression models were used to explore the effect of PCOS status on TSH cut-off values and SCH, respectively. The 95 percentiles of TSH were not significantly different in the PCOS group compared to control ones (6.12 and 6.56 microU/mL, respectively). There was no statistically significant association between PCOS status and SCH (OR adjusted: 1.40; 95%CI: 0.79, 2.50; *p* = 0.2). The prevalence of SCH and the upper reference limit of TSH were not significantly different in PCOS and controls. Investigation of SCH in women with PCOS might be questionable.

## 1. Introduction

Polycystic ovary syndrome (PCOS) was first described by Stein and Leventhal in 1935 in women with amenorrheic morphology and clinical evidence of an androgen increase [1]. The combination of hyperandrogenism and anovulation remained the pillar of PCOS definition for 50 years [2]. Afterward, the Rotterdam criteria for PCOS diagnosis was introduced in 2003, which is defined by the presence of any two of the three following manifestations: Oligo/anovulation, polycystic ovaries, clinical and/or biochemical hyperandrogenism [3]. Exclusion of other pathologies mimicking PCOS, including hyperprolactinemia and any endocrinological disorder related to the thyroid, pituitary, and adrenal gland, is a mandatory component of PCOS diagnosis across all the defined criteria over time [4].

Hypothyroidism is one of the most common endocrinopathies in women and the most common thyroid disorder in women of reproductive age [5]. Thyroid hormones are necessary for female reproduction and directly affect the development and metabolism of ovaries and the uterine [6,7]. Thus, hypothyroidism in women is usually accompanied by reproduction disorders such as delay in puberty, ovarian cysts, anovulation, and menstrual irregularities [8]. 

PCOS patients may exhibit features of metabolic abnormalities such as insulin resistance, obesity, and dyslipidemia [9]. Hypothyroidism has also been shown to reduce glucose production and consumption, leading to insulin resistance. Deficiency in thyroid hormones can also be associated with weight gain and excess body mass, hyperlipidemia, decreased sex hormone-binding globulin (SHBG) levels and increased conversion of androstenedione to testosterone [10]. Moreover, hypothyroidism may also affect gonadal function, leading to anovulatory cycles [11]. Due to the association between insulin residence and reproductive disorder with both hypothyroidism and PCOS, patients with overt hypothyroidism are excluded from being diagnosed with PCOS [12]. The questions now are whether PCOS influences the cut-off value of thyroid stimulating hormone (TSH) and whether subclinical hypothyroidism (SCH), which is defined by TSH levels above the upper limit of the normal range accompanied by normal levels of free thyroxine, is more common in women with PCOS [13]. 

A meta-analysis based on six studies reported a higher prevalence of SCH among PCOS patients; however, the majority of the studies included were clinical-based, which may lead to selection bias [14]. Only one study reported that SCH does not increase the risk of PCOS in obese women of reproductive age after adjustment for potential confounders [15]. To date, the crosstalk between PCOS and subclinical hypothyroidism is still a topic of debate, and our knowledge of the interplay between PCOS and TSH cut-off values is limited due to the limited number of population-based studies.

In the present study, we aimed to assess if there is a difference between PCOS and the control group in terms of the upper reference limit of TSH. We also aimed to identify the prevalence of SCH in women with PCOS compared to controls in a population-based study.

## 2. Materials and Methods

This study was conducted on data collected in the Iranian PCOS prevalence study [16] and the Khuzestan PCOS prevalence study [17]. The details of these studies have been published before. In brief, the Iranian PCOS prevalence study was conducted on 1126 women recruited among reproductive-aged women of four randomly selected provinces of different geographic regions of Iran; and the Khuzestan PCOS study was conducted on 646 women aged 18–45 years, living in urban areas of Khouzestan province. In brief, a standard questionnaire was completed during face-to-face interviews by trained midwives under the supervision of a gynecologist. All participants underwent clinical examinations, and an overnight fasting venous blood sample was obtained from each subject (regardless of PCOS status) on the second or third day of their spontaneous or progesterone-induced menstrual cycles, at least 2 h after wake-up. All the subjects underwent a transvaginal scan or transabdominal ultrasonography of the ovaries, performed using the 3.5-MHz transabdominal and 5-MHz transvaginal transducer, respectively. An ultrasound was performed on the same day as blood samples were collected. 

For the purpose of the present study, we excluded all menopausal women (natural/surgical) (*n*= 49), pregnant or lactating women (*n* = 52), those with hyperprolactinemia (*n* = 32), women with overt thyroid dysfunction (*n* = 24), women presenting only one feature of PCOS Rotterdam criteria, including women with only hyperandrogenism (clinical/biochemical) (*n*= 337), women with only anovulation (*n* = 112), women with only polycystic ovarian morphology (*n* = 133), smokers (*n* = 27), those taking medications known to affect hormonal or metabolic parameters (*n* = 12), and those with missing data (*n* = 76). No cases of Cushing’s syndrome, congenital adrenal hyperplasia, or virilizing tumors were diagnosed using appropriate tests. We further excluded all those participants aged <18 or >45 (*n* = 44) and those with TSH value >= 10 (*n* = 23)

The remaining participants were categorized into two groups:PCOS group: Women diagnosed with PCOS based on the Rotterdam criteria (*n* = 207).Control: Those without any feature of PCOS; eumenorrheic not hirsute women without polycystic ovary morphology (PCOM) (*n* = 644).

### 2.1. Definitions

Rotterdam criteria were used for PCOS diagnosis, defined as any two of the following manifestations: Oligo/anovulation, polycystic ovaries, and clinical and/or biochemical hyperandrogenism [3]. Thyroid dysfunction was defined based on TSH (0.32–5.06 µ/L) and free thyroxin (FT4) (0.91–1.55 ng/dL) [18].

### 2.2. Measurements

All hormonal assessment is presented in Table 1. All the laboratory measurements were carried out at the same laboratory (Endocrine Research Center). Free androgen index (FAI) was calculated using the formula [TT (nmol/L) × 100/SHBG (nmol/L)]. 

### 2.3. Statistical Analysis

Continuous variables were presented as mean (standard deviation (SD) if normally distributed, median (interquartile range (IQR)) if not, and categorical variables were presented as numbers (%) among the normal and PCOS groups. To explore the effect of PCOS status on subclinical hypothyroidism, a logistic regression model was used and odds ratios; 95% confidence intervals were also estimated. Both unadjusted and adjusted models were reported. Furthermore, because the TSH levels were not normally distributed, quantile regression model was used to assess the effect of PCOS on the TSH. Quantile regression is a flexible and robust methodology in which coefficients reveal the effect of a unit change in the covariate on the quantiles of the response distribution [19]. Both unadjusted and adjusted models were applied, and adjusted variables were defined based on age, BMI and number of pregnancies which were potential available confounders. Moreover, boxplot of TSH was drawn based on PCOS status. Statistical analysis was performed using software package STATA (version 12; STATA Inc., College Station, TX, USA); significance level was set at *p* < 0.05, and CI as 95%.

## 3. Results

Flowchart of the study is presented in Figure 1. Table 2 illustrates the clinical and endocrine characteristics of women participating in this study according to their PCOS status. Women in the control group were significantly older than women with PCOS (34.1 (7.3) vs. 30.7 (7.5), *p* < 0.001). No statistically significant differences were reported in the anthropometric measurements, including BMI (body mass index), waist circumference, hip circumference, WHR (waist-hip ratio), and WHtR (waist-to-height ratios) between PCOS and the control group (*p* > 0.05). The median (inter quartile range) of TSH in the control and PCOS groups were 2.1 (1.5–3.4) and 2.0 (1.4–3.4) (microU/mL), respectively (*p* = 0.2). SCH was observed in 24 (11.6) of PCOS women and 65 (10.1) of their control counterparts (*p* = 0.5). Table 3 present the clinical and endocrine characteristics of those with SCH in PCOS and control ones.

The results of the non-adjusted and adjusted logistic regression models are presented in Table 4; there was no association between PCOS status and SCH (OR adjusted: 1.40; 95%CI: 0.79, 2.50; *p* = 0.2).

Figure 2 demonstrates the Box plots for TSH (microU/mL) among the study participants with respect to their status of PCOS. The 95 percentiles of TSH were not significantly different in PCOS group in comparison to control ones; 6.12 and 6.56 (microU/mL), respectively. Table 5 presented the results of the quantile regression model; no significant difference was observed between the two groups in any quintile range, except for the median in the adjusted model. After adjusting for age, BMI, and parity, the median of TSH in the PCOS group was 0.2 (microU/mL) lower than their control counterpart (−0.16; 95%CI: −0.32, −0.01; *p* = 0.04).

## 4. Discussion

In the present population-based study, using quantile regression models, we found no significant difference in any quintile range of TSH between women with PCOS and controls after adjustment for potential confounders, except for the median of TSH in adjusted model, which was 0.16 microU/mL lower in PCOS group; the upper reference limit of TSH (95 percentile) was not significantly different in PCOS group in comparison to control ones. There was also no significant difference in the prevalence of subclinical hypothyroidism between women with PCOS and controls after adjusting for potential confounders.

Several assumptions may explain the association between thyroid hormones and PCOS; this syndrome is associated with impaired pulsatile LH secretion and decreased SHBG through increased TRH secretion and a subsequent increase in prolactin secretion, resulting in an increase in testosterone levels [12,20]. On the other hand, glucose production and consumption are reduced by hypothyroidism, leading to insulin resistance [9]. Deficiency in thyroid hormones can also be associated with weight gain and excess body mass, dyslipidemia, decreased sex hormone-binding globulin (SHBG) levels, and increased conversion of androstenedione to testosterone [10]; these disturbances are also observed in women with PCOS [9]. It is well-documented that overt hypothyroidism affect gonadal function, leading to anovulatory cycles [11]. As a consequence, thyroid dysfunction and PCOS may manifest with the same symptoms, including menstrual irregularity, ovulation disorders, infertility, endometrial thickness, and polycystic ovary appearance [12,20]. Therefore, excluding overt hypothyroidism in making the diagnosis of PCOS is mandatory since PCOS manifestations may be explained by thyroid dysfunction rather than suffering from the simultaneous condition of both PCOS and hypothyroidism [21,22].

Despite this solid evidence for overt hypothyroidism, the association between PCOS and subclinical hypothyroidism is still a topic of debate. Our study indicates that SCH prevalence is comparable in both groups of PCOS and controls. Several studies explored this association and reported inconclusive results [14,15]. While the majority of clinical-based studies reported a higher prevalence of SCH in PCOS women [23,24,25,26,27,28], Bingjie Zhang et al. [15], in a study on 534 obese women of reproductive age, suggested that SCH does not increase the risk of PCOS after adjusting for confounding factors. In 2018, a meta-analysis of six clinical studies by Xiaohong Ding et al. [14] on 692 PCOS patients compared to 540 controls demonstrated a significant 2.87 combined odd ratio of SCH risk for PCOS patients compared to controls, which increased to 3.59 when limiting TSH cut-off to ≥4 mIU/L. Several limitations could be attributed to the findings of this study: First, all the PCOS patients in the studies included in this meta-analysis were recruited from tertiary hospitals or infertility clinics. As a result, the PCOS group subjects in this study were selected from patients with more severe PCOS phenotypes. Thus, the findings may not be generalized to all spectrum of PCOS phenotypes including milder ones. Second, the findings may be biased due to different studies’ heterogeneity in PCOS and SCH definitions. Finally, the sample size is still too small to avoid a random error. 

The relationship between PCOS and thyroid dysfunction may also be partly explained by the possible autoimmunity pathogenesis of both conditions. Autoimmune thyroiditis is considered the main cause of SCH [29]. Similarly, autoimmunity also plays a role in the pathogenesis of PCOS through the possible discordance between estrogen and progesterone. Estrogen is an immune system stimulator modulated by progesterone [30]. In women with PCOS, anovulatory cycles lead to low progesterone levels. Furthermore, it increases the ratio of estrogen to progesterone, which may compromise the immune system, leading to an increase in autoimmunity [31,32]. A meta-analysis of 13 clinical-based studies carried out by Mírian Romitti et al. [33] of a total of 1210 PCOS patients compared to the control group concluded that autoimmune thyroid disease is more frequent among women diagnosed with PCOS. Unfortunately, we did not measure autoimmune thyroid parameters such as anti-TPO antibodies, and a thyroid ultrasound was not performed on the study participants to compare the prevalence of autoimmune thyroiditis in the two groups. Thus, we could not investigate the cause of SCH from the perspective of autoimmunity. However, since Iran is an iodine-deficient area [34,35], it seems that the most common cause of thyroid dysfunction in Iran is iodine deficiency [36,37].

In the present study, we found no significant difference in any quintile range of TSH between women with PCOS and controls after adjustment for potential confounders; the presence of PCOS neither impacts the level of TSH values nor the 95th percentile that mainly used as the threshold of TSH. Using quantile regression analysis has the advantage of understanding the relationships between PCOS and SCH outside of the data’s mean, making it useful in understanding outcomes that are non-normally distributed and that have nonlinear relationships with predictor variables [38]. The majority of available studies have investigated the mean value of TSH between women with PCOS and controls with inconclusive results [33]. While six studies demonstrated higher TSH values in PCOS women compared to control [23,25,26,28,31,39], other studies revealed comparable values of TSH for both PCOS and control groups [40,41,42,43,44] 

In addition to the design of our study (population-based), different results of our present study in comparison to those that reported higher prevalence of SCH or higher TSH levels may be partly explained by the difference in assessing the confounding effect of factors such as BMI and parity on their results since higher levels of BMI have been directly connected to an increase in TSH values [45,46], and it has been shown that the number of parities has a positive association with autoimmune thyroiditis [47]. For example, in a cross-sectional study by Janssen et al. [31]. on 175 patients with PCOS and 168 age-matched controls, TSH values for the PCOS group were reported to be significantly higher, which could be biased due to the confounding effect of BMI on TSH values. The high prevalence of SCH in the current study may be explained by fast iodine supplementation, and vitamin D deficiency. It has been shown that increases in iodine intake in people living in iodine-deficient areas is associated with higher levels of TSH and thyroid autoimmunity [48]. Additionally, studies have shown that vitamin D deficiency is directly related to the increase in autoimmune thyroid diseases [49]. Our study was carried out on the Iranian population, a region with a very high prevalence of vitamin D deficiency [50], causing a possible increase in autoimmune diseases. On the other hand, the higher consumption of vitamin D supplements by women diagnosed with PCOS [51] might be a rationale for reducing autoimmune thyroid disease and the prevalence of SCH among these patients.

The strength of our study is that it is the first population-based study with high statistical power in which we have compared the prevalence of subclinical hypothyroidism in women with PCOS and controls. Using the quantile regression approach is the second strength of the present study; this approach quantifies the association of explanatory variables with a conditional quantile of a dependent variable without assuming any specific conditional distribution; this method allows for the understanding of the relationships between variables outside of the mean of the data, making it useful in understanding outcomes that are non-normally distributed and that have nonlinear relationships with predictor variables [52]. In addition, our study subjects were in a broader age range than previous studies so that we could compare TSH levels in the full range of reproductive ages between PCOS and control groups. Moreover, an ultrasound was used to diagnose PCOS in all study subjects. As a result, the possibility of misclassifying PCOS subjects with mild and subclinical phenotypes was minimal. Finally, in our study, all the laboratory measurements were conducted in the same laboratory with the same method. As a result, the intra-assay variability in our data is likely to be negligible. However, there are some limitations worth mentioning. First, we did not measure autoimmune thyroid factors such as anti-TPO antibodies, and a thyroid ultrasound was not performed on the study participants to compare the prevalence of autoimmune thyroiditis in the two groups. Second, we did not measure the vitamin D levels of our study participants to assess our hypothesis of the effect of vitamin D deficiency on our results. Our data do not have adequate power to conduct the subgroup analysis according to the PCOS phenotypes. 

## 5. Conclusions

In a population-based study, we demonstrated that the prevalence of SCH does not differ between PCOS and control groups. Moreover, the threshold of TSH values is comparable in both groups. This finding adds to our knowledge about the relationship between hypothyroidism and PCOS. Additional comprehensive population-based studies with a sufficiently large sample size and a more detailed assessment of thyroid status are needed. Investigation of SCH in women with PCOS might be questionable.

## Figures and Tables

**Figure 1 diagnostics-13-00316-f001:**
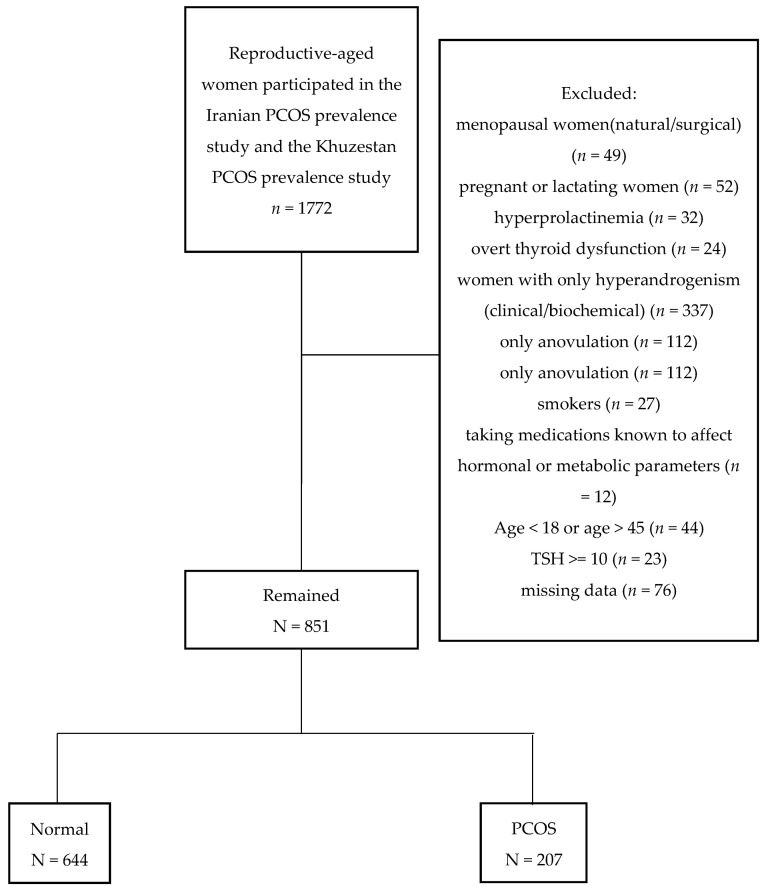
Flowchart of the study.

**Figure 2 diagnostics-13-00316-f002:**
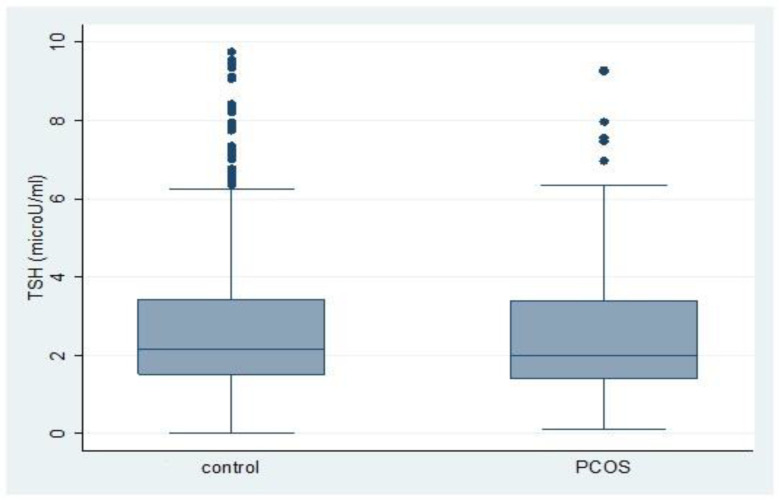
Box plots for TSH (microU/mL) among the study participants with respect to their status of PCOS. Note: Boxes show 1st, 2nd (median), and 3rd quartile of TSH for each group, respectively. Outliers (the values outside of one and a half interquartile range) have been shown by dots.

**Table 1 diagnostics-13-00316-t001:** Laboratory assessments.

Hormone	Measurement Method	Kit & Instrument	Intra-Assay	Inter-Assay
Total testosterone (TT)	enzyme immunoassay (EIA)	Diagnostic biochem Canada Co., London, ON, Canada	1.7%	2.3%
Dehydroepiandrosterone sulfate (DHEAS)	enzyme immunoassay (EIA)	DRG Instruments, GmbH, Marburg, Germany	1.9%	2.5%
Sex hormone-binding globulin (SHBG)	immunoenzymometric assay (IEMA)	Diagnostic biochem Canada Co., London, ON, Canada	0.8%	2.4%
Luteinizing hormone (LH)	immunoradiometric assay (IRMA)	commercial kits (Izotop, Budapest, Hungary) and a Dream Gamma- 10 gamma counter (Shin Jin Medics Inc., Koyang, Korea)	1.6%	4.2%
follicle-stimulating hormone (FSH)			1.4%	2%
thyroid-stimulating hormone (TSH) *			2.1%	3.1%
FT4 *	electrochemiluminescence immunoassay (ECLIA)	commercial kits on the Cobas e411 analyzer (Roche Diagnostics GmbH, Mannheim, Germany)	1.3%	3.3%

* The sensitivity of FT4 and TSH measurements were 0.023 ng/dl and 0.005 mU/L, respectively.

**Table 2 diagnostics-13-00316-t002:** Clinical and endocrine characteristics of the PCOS cases and controls.

Variable	Total (*n* = 851)	Control (*n* = 644)	PCOS (*n* = 207)	*p*-Value
Age ^a^ (years)	33.3 (7.5)	34.1 (7.3)	30.7 (7.5)	**<0.001**
BMI ^a^ (kg/m^2^)	26.7 (5.2)	26.7 (5.1)	26.6 (5.5)	0.8
Number of pregnancy ^a^	2.8 (1.6)	2.9 (1.6)	2.5 (1.5)	**0.01**
Number of delivery ^a^	2.5 (1.4)	2.5 (1.4)	2.1 (1.3)	**0.001**
Number of abortion ^a^	0.4 (0.7)	0.4 (0.7)	0.4 (0.6)	0.9
WC ^a^ (cm)	83.2 (11.8)	83.1 (11.7)	83.2 (12.3)	0.9
Height ^a^ (cm)	158.9 (6.2)	158.9 (6.2)	159.3 (6.1)	0.4
Weight ^a^ (kg)	67.3 (13.0)	67.3 (12.8)	67.4 (13.7)	0.9
Hip ^a^ (cm)	102.9 (12.2)	102.9 (11.9)	102.9 (13.3)	0.9
WHR ^a^	0.81 (0.09)	0.81 (0.09)	0.81 (0.07)	0.9
WHtR ^a^	0.52 (0.08)	0.52 (0.08)	0.52 (0.08)	0.8
FAI ^b^	1.9 (0.8,3.6)	1.7 (0.7,3.2)	3.3 (1.5,5.4)	**<0.001 ^d^**
FSH ^b^ (microU/mL)	7.6 (5.7,9.8)	8.0 (5.9,10.1)	8.9 (6.9,10.4)	**<0.001 ^d^**
LH ^b^ (microU/mL)	4.7 (3.4–6.5)	4.6 (3.4–6.4)	4.9 (3.3–7.0)	0.2
LH/FSH ^b^	0.6 (0.42–0.85)	0.58 (0.40–0.81)	0.69 (0.48–0.98)	**<0.001 ^d^**
SHBG ^b^ (nmol/L)	56.4 (42.7–81.9)	59.4 (43.8–85.6)	50.5 (40.8–67.5)	**0.001 ^d^**
DHEAS ^b^ (microg/dL)	138.9 (73.6–195.6)	118.9 (65.6–186.7)	179.4 (119.3–221.0)	**<0.001 ^d^**
Testosterone ^b^ (nmol/L)	0.35 (0.14–0.61)	0.31 (0.12–0.56)	0.49 (0.24–0.79)	**<0.001 ^d^**
TSH ^b^ (microU/mL)	2.1 (1.5–3.4)	2.1 (1.5–3.4)	2.0 (1.4–3.4)	0.2
SCH ^c^	89 (10.5)	65 (10.1)	24 (11.6)	0.5

^a^ Values are presented as mean (SD), ^b^ values are expressed as median (Inter Quartile Range), ^c^ data shown as number (percentage). Analyzed using independent samples *t*-test for superscripts a, Mann–Whitney U test for superscripts b and Pearson’s test for superscripts c. PCOS polycystic ovarian syndrome, BMI body mass index, WC waist circumference, WHR waist–hip ratio, WHtR waist-to-height ratio, FAI free androgen index, FSH follicle stimulating hormone, LH luteinizing hormone, FSH follicle stimulating hormone, SHBG sex hormone-binding globulin, DHEAS dehydroepiandrosterone sulfate, TSH thyroid stimulating hormone, SCH subclinical hypothyroidism. ^d^ *p* < 0.001 versus controls after adjusting for age, BMI and parity.

**Table 3 diagnostics-13-00316-t003:** Clinical and endocrine characteristics of women with SCH according to the PCOS status.

Variable	SCH PCOS (*n* = 24)	SCH Control (*n* = 65)	*p*-Value
Age ^a^ (years)	32.3 (8.1)	35.1 (7.5)	0.1
BMI ^a^ (kg/m^2^)	28.0 (5.6)	27.8 (4.5)	0.8
Number of pregnancy ^a^	2.7 (1.7)	2.8 (1.4)	0.8
Number of delivery ^a^	2.1 (1.4)	2.5 (1.3)	0.3
Number of abortion ^a^	0.5 (0.6)	0.2 (0.4)	0.07
WC ^a^ (cm)	87.9 (11.8)	86.3 (9.6)	0.5
Height ^a^ (cm)	158.9 (4.7)	158.6 (5.7)	0.9
Weight ^a^ (kg)	70.6 (13.7)	69.9 (10.7)	0.8
Hip ^a^ (cm)	107.7 (10.6)	105.2 (9.7)	0.3
WHR ^a^	0.81 (0.07)	0.82 (0.06)	0.4
WHtR ^a^	0.55 (0.08)	0.54 (0.06)	0.6
FAI ^b^	4.9 (2.4–7.0)	2.7 (1.2–3.8)	**0.004 ^d^**
FSH ^b^ (microU/mL)	6.7 (5.5–9.0)	6.5 (5.2–9.6)	0.9
LH ^b^ (microU/mL)	4.3 (3.5–7.9)	4.5 (3.5–6.4)	0.5
LH/FSH ^b^	0.6 (0.5–1.3)	0.6 (0.5–0.8)	0.5
SHBG ^b^ (nmol/L)	44.1 (33.2–54.8)	60.1 (49.9–89.3)	**0.001 ^d^**
DHEAS ^b^ (microg/dL)	191.9 (148.0–221.5)	136.1 (55.8–214.8)	**0.01 ^f^**
Testosterone ^b^ (nmol/L)	0.6 (0.3–0.8)	0.5 (0.2–0.7)	0.08

^a^ Values are presented as mean (SD), ^b^ values are expressed as median (Interquartile Range), ^c^ data shown as number (percentage). Analyzed using independent samples *t*-test for superscripts a, Mann–Whitney U test for superscripts b and Pearson’s test for superscripts c. BMI body mass index, WC waist circumference, WHR waist–hip ratio, WHtR waist-to-height ratio, FAI free androgen index, FSH follicle stimulating hormone, LH luteinizing hormone, FSH follicle stimulating hormone, SHBG sex hormone-binding globulin, DHEAS dehydroepiandrosterone sulfate, TSH thyroid stimulating hormone, SCH subclinical hypothyroidism, PCOS polycystic ovarian syndrome. ^d^ *p* < 0.001 versus controls after adjusting for age, BMI and parity. ^f^ *p* < 0.05 versus controls after adjusting for age, BMI and parity.

**Table 4 diagnostics-13-00316-t004:** Logistic regression model analysis for exploring the effect of PCOS on SCH before and after adjustment for potential confounders.

Model	Variable	OR	95% CI	*p*-Value
Unadjusted model	PCOS	1.17	0.71, 1.92	0.5
Adjusted model *	PCOS	1.40	0.79, 2.50	0.2
	Age (year)	1.02	0.97, 1.07	0.4
	BMI (kg/m^2^)	1.04	0.99, 1.09	0.1
	parity	0.90	0.74, 1.09	0.3

* Model adjusted for age, BMI, and parity; reference group: control, PCOS polycystic ovarian syndrome, SCH subclinical hypothyroidism, BMI body mass index.

**Table 5 diagnostics-13-00316-t005:** Quantile regression model for exploring the effect of the PCOS status on TSH percentiles.

Model	5th Centile		25th Centile		Median		75th Centile		95th Centile	
	Coef. 95% CI	*p*-Value	Coef. 95% CI	*p*-Value	Coef. 95% CI	*p*-Value	Coef. 95% CI	*p*-Value	Coef. 95% CI	*p*-Value
Unadjusted	−0.19 −0.47, 0.09	0.2	−0.10 −0.29,0.09	0.3	−0.16 −0.39, 0.07	0.2	−0.04 −0.84, 0.76	0.9	−0.44 −1.25, 0.37	0.3
Adjusted *	−0.18 −0.44, 0.06	0.1	−0.07 −0.26, 0.11	0.4	−0.16 −0.32, −0.01	**0.04**	0.05 −0.65, 0.77	0.9	−0.24 −1.55, 1.07	0.7

* Model adjusted for age, BMI, and parity; reference group: control, PCOS polycystic ovarian syndrome, SCH subclinical hypothyroidism, BMI body mass index before and after adjustment for age, BMI and parity.

## Data Availability

Not applicable.

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
