# Peer review of "Polycystic Ovary Syndrome, Subclinical Hypothyroidism, the Cut-Off Value of Thyroid Stimulating Hormone; Is There a Link? Findings of a Population-Based Study"

_diagnostics, 2023, doi:10.3390/diagnostics13020316_

Round 1

Reviewer 1 Report

The study investigates the possible correlation between PCOS and subclinical hypothyroidism.

The study is well designed, performed and written.

Author Response

Point 1: The study investigates the possible correlation between PCOS and subclinical hypothyroidism.

The study is well designed, performed and written.

Response 1: Thank you for your acknowledgment of our study and positive feedback.

Reviewer 2 Report

Reviewer Comments to Author

The manuscript entitled “Polycystic ovary syndrome, subclinical hypothyroidism, the cut-off value of thyroid stimulating hormone; is there a link? Findings of a population-based study” researched the relationship between PCOS and subclinical hypothyroidism (SCH) . Overall, the article puts forward new ideas, but there are several issues as described below:

1.In the introduction part: Due to the association of insulin residence and reproductive disorder with both hypothyroidism and PCOs,” please check the PCOS.

2. Materials and Methods : “2- Control: Those without any feature of PCOS; eumenorrheic not hirsute women without PCOM (n=644).” What is PCOM? 

“Thyroid dysfunction was defined based on TSH (0.32–5.06 µ/L) and free thyroxin (FT4) (0.91–1.55 ng/dL) [18]. “ This sentence should be placed in suit place.

3.The section of the article is not clear, pay attention to the format of "Measurements", you can add a subtitle to "Measurements" and "Statistical analysis".

4. Below the Table 1, PCOS polycystic ovarian syndrome BMI body mass index, this sentence shouldbe added one ,.

5. There are some problems with the article typesetting.

6.In the discussion part: , Bingjie Zhang et al [15]., in a study on 534 obese women of reproductive age,” please note the punctuation marks.

7. The discussion section is too lengthy, focusing on the connection between PCOS and subclinical hypothyroidism (SCH) and the value of the results of this study. I hope the author can think more.

Author Response

Thank you for your review of our manuscript. We appreciate the time and effort you have taken to provide us with your comments and suggestions.

We have carefully considered your feedback and responded to your comments.

Response to Reviewer 2 Comments

Point 1: In the introduction part: “Due to the association of insulin residence and reproductive disorder with both hypothyroidism and PCOs,” please check the PCOS.

Response 1: Corrected as suggested.

Point 2: Materials and Methods : “2- Control: Those without any feature of PCOS; eumenorrheic not hirsute women without PCOM (n=644).” What is PCOM?

“Thyroid dysfunction was defined based on TSH (0.32–5.06 µ/L) and free thyroxin (FT4) (0.91–1.55 ng/dL) [18]. “ This sentence should be placed in suit place.

Response 2: PCOS is defined as polycystic ovary morphology, and in the revised manuscript, it has been corrected as suggested.

The above sentence has been rewritten as you suggested in a subsection titled "Definitions".

Point 3: The section of the article is not clear, pay attention to the format of "Measurements", you can add a subtitle to "Measurements" and "Statistical analysis".

Response 3: Corrected as suggested.

Point 4: Below the Table 1, “PCOS polycystic ovarian syndrome BMI body mass index,” this sentence shouldbe added one “,”.

Response 4: Corrected as suggested.

Point 5: There are some problems with the article typesetting.

Response 5: Corrected as suggested.

Point 6: In the discussion part: “, Bingjie Zhang et al [15]., in a study on 534 obese women of reproductive age,” please note the punctuation marks

Response 6: Corrected as suggested.

Point 7: The discussion section is too lengthy, focusing on the connection between PCOS and subclinical hypothyroidism (SCH) and the value of the results of this study. I hope the author can think more.

Response 7: In the manuscript's revision, the article's discussion part was tried to be as short as possible. However, due to the subject's controversial nature, the need for a pathophysiological explanation is felt. We hope that the changes we have made will address your concern.

Reviewer 3 Report

Clinical problems concerning patients with PCOS spectrum disorders are hot-topic and at glance it seems that for longer period will remain in our interest. We are highly interested in solving problems associated with this multidirectional hormonal imbalance. 

I found submitted manuscript interesting and worth to be published. I have rather small remarks and found flaws are easy to be fixed. 

Do the readers have opportunity to get familiar with questionnaire? It should be placed in the supplementary data.

Measurements. Should be placed in the table. It would be much easier for the reader to get familiar -> methods, ranges, equipment, etc.

In the table I as well in table II the Authors mixed clinical characteristics of the patients which should be placed in the M&M section with measurable and quantifiable data/findings, e.g. blood results...

Figure 2 is not clearly explained. It is non-informative. What is shown by dots what by boxes?

In the perspective general knowledge as well cited papers, it's pity that in the presented groups leves of vit. D were not analyzed.

References with epidemiological studies older then 10 years shouldn't be cited. Does not implement any important/new data. There are really fresh papers which may be placed in the ref. list.

Author Response

Thank you for your review of our manuscript. We appreciate the time and effort you have taken to provide us with your comments and suggestions.

We have carefully considered your feedback and responded to your comments.

Response to Reviewer 3 Comments

Point 1: Do the readers have opportunity to get familiar with questionnaire? It should be placed in the supplementary data.

Response 1: Including study questionnaires in the article may be possible, but it will require that the questionnaires be translated into English first.

Point 2: Measurements. Should be placed in the table. It would be much easier for the reader to get familiar -> methods, ranges, equipment, etc.

Response 2: As you suggested, in the revised manuscript, all hormonal assessment is presented in table 1.

Point 3: In the table I as well in table II the Authors mixed clinical characteristics of the patients which should be placed in the M&M section with measurable and quantifiable data/findings, e.g. blood results...

Response 3: According to your recommendation in the previous point, methods of assessment of all hormones are presented in table 1 in the method section of the article. The results of these hormonal measurements are presented in Tables 2 and 3 in the result section of the article. We hope that the changes we have made will address your concern.

Point 4: Figure 2 is not clearly explained. It is non-informative. What is shown by dots what by boxes?

Response 4: Figure 2 demonstrates the Box plots for TSH (microU/ml) among the study participants with respect to their status of PCOS. A boxplot displays the median, the quartiles, adjacent values, the range of values covered by the data and any outliers which may be present. It also allows a visual of lack of symmetry. Boxes show each group's first, second (median), and third quartile of TSH. For example, these amounts for TSH in the control group are 1.45, 2.14, and 3.43, respectively. For those with PCOS, the box shows the number of 1.41, 1.98, and 3.39 as first, median and third quartile, respectively.

Outliers, which dots have shown, are the values outside of one and a half IQR (Q3-Q1) of either the first or third quartiles. In the control group, Q3+1.5 IQR is equal to 6.31, and there are 36 women with TSH larger than 6.31 in this group.

This amount for PCOS women is equal to 6.36, and only 6 PCOS women had TSH greater than 6.36. Moreover, we can observe maximum values in PCOS, and the control group are 9.28 and 9.75, respectively.

Figure 2 is revised accordingly.

Point 5: In the perspective general knowledge as well cited papers, it's pity that in the presented groups levels of vit. D were not analyzed.

Response 5: Unfortunately, this measurement has not been considered for this study. We acknowledged it as a limitation.

Point 6: References with epidemiological studies older then 10 years shouldn't be cited. Does not implement any important/new data. There are really fresh papers which may be placed in the ref. list.

Response 6: References were revised, and epidemiologic studies published more than ten years were replaced with newer studies. While a small number of older studies on the underlying mechanisms are still referenced, the majority of current research is based on more recent findings.
